# Detecting the Surface Signature of Riverine and Effluent Plumes along the Bulgarian Black Sea Coast Using Satellite Data

Irina Gancheva [1,2,*], Elisaveta Peneva [2] and Violeta Slabakova [3]

1 European Space Agency, Centre for Earth Observation, Largo Galileo Galilei 1, 00044 Frascati, RM, Italy
2 Department of Meteorology and Geophysics, Faculty of Physics, Sofia University "St. Kliment Ohridski",
  5 James Bourchier Blvd, 1164 Sofia, Bulgaria; elfa@phys.uni-sofia.bg
3 Department of Ocean Technologies, Institute of Oceanology, Bulgarian Academy of Sciences,
  First May Street 40, 9001 Varna, Bulgaria; v.slabakova@io-bas.bg
* Correspondence: irina.gancheva@phys.uni-sofia.bg

**Abstract:** The clear and reliable detection of effluent plumes using satellite data is especially challenging. The surface signature of such events is of a small scale; it shows a complex interaction with the local environment and depends greatly on the effluent and marine water constitution. In the context of remote sensing techniques for detecting treated wastewater discharges, we study the surface signature of small river plumes, as they share specific characteristics, such as higher turbidity levels and increased nutrient concentration, and are fresh compared to the salty marine water. The Bulgarian Black Sea zone proves to be a challenging study area, with its optically complex waters and positive freshwater balance. Additionally, the Bulgarian Black Sea coast is a known tourist destination with an increased seasonal load; thus, the problem of the identification of wastewater discharges is a topical issue. In this study, we analyze the absorption components of the Inherent Optical Properties (IOPs) for 84 study points that are located at outfall discharging areas, river estuaries and at different distances from the shoreline, reaching the open sea area at a bottom depth of more than 2000 m. The calculations of IOPs take into account all available Sentinel 2 cloudless acquisitions for three years from 2017 until 2019 and are performed using the Case-2 Regional CoastColour (C2RCC) processor, implemented in the Sentinel Application Platform (SNAP). The predominant absorber for each study area and its temporal variation is determined, deriving the specific characteristics of the different areas and tracking their seasonal and annual course. Optical data from the Galata AERONET-OC site are used for validating the absorption coefficient of phytoplankton pigment. A conclusion regarding the possibility of distinguishing riverine, marine and coastal water is derived. The study provides a sound basis for estimating the advantages and drawbacks of optical satellite data for tracking the extent of effluent and fluvial plumes with unknown concentrations of optically significant seawater constituents.

**Keywords:** wastewater detection; effluent detection; riverine plume detection; Inherent Optical Properties; C2RCC processor; Black Sea; ocean remote sensing

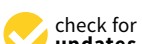



## 1. Introduction

Effluent water is discharged directly into the riverine or marine environment, and, in the case of insufficient treatment, this can have a significant impact on the ecosystem, alter the state of biological communities or impact human health. Small water basins and semi-enclosed seas are especially vulnerable to wastewater pollution due to their limited water exchange with the global oceans and locally adjusted ecosystems. Independent and continuous monitoring of such discharges is of great importance for coastal zone management, sustainable development and the blue growth of coastal regions. In this regard, satellite remote sensing offers an irreplaceable information source.

Poorly treated or completely untreated wastewater (WW) can pose a serious threat to the aquatic environmental system. Prolonged exposure to such contamination might alter the biogeochemical composition of the water and have long-term consequences [1,2]. Studies of the long-term effect of effluent discharge on the marine environment reveal that prolonged discharge of poorly treated effluents alters the biological impact on the benthic communities, leading to degradation [2]. Nevertheless, even after mechanical and biological treatment, effluent discharges can still contain contaminants, such as metals [3], nutrients, oil droplets or harmful chemicals [4].

An extensive legal framework has been developed regarding the monitoring and regulation of the procedures for WW treatment and discharge. The World Health Organization revised its guidelines for WW management in 2006 [5], and the European Union has introduced a Water Framework Directive, with sub-directives explicitly regulating urban WW quality [6,7], stressing the importance of adequate WW treatment and its potential hazards.

At the same time, current studies on the quality of treated WW effluent discharges in Bulgaria reveal concerning results about the concentrations of various physiochemical and microbiological wastewater parameters [8,9].

Previous studies on the potential of satellite data for detecting effluent and stormwater plumes have demonstrated the capabilities of multispectral, radar and thermal sensors for these purposes [10–16]. Gierach et al. [10] conducted a multi-sensor analysis of two diversion events in major wastewater treatment plants in Southern California, detecting the wastewater surface plume and demonstrating the advantages and constraints of different data sources. DiGiacomo et al. [12] used Synthetic Aperture Radar (SAR) data to detect the surface signature of stormwater runoff plumes, municipal wastewater plumes and natural hydrocarbon seeps in the Southern California Bight by analyzing the sea surface roughness. Marmorino et al. [17] used airborne hyperspectral and infrared imagery for detecting wastewater by the increase in Colored Dissolved Organic Matter (CDOM) plus detrital material and decreased sea surface temperature. In a recent study, Ayad et al. [16] used the differences in the spectral features in the visible to near-infrared (NIR) bands, together with in situ measurements, to investigate the Tijuana River mouth in California. With this approach, they could distinguish between four key plume categories: stormwater, wastewater, open ocean/no plume and mixed. Currently, this is the only published work that successfully distinguishes between stormwater plumes and treated wastewater plumes.

There are several studies dedicated to river plume extent and stormwater runoff mapping using airborne and satellite data [18–21]. The potential of the Sea-Viewing Wide Field-of-View (SeaWiFS) ocean color sensor for evaluating the effect of stormwater runoff on phytoplankton blooms was demonstrated by Nezlin et al. [22]. Estimation of the extent of polluted stormwater plumes using a combined near-infrared/shortwave-infrared (NIR/SWIR) atmospheric correction method for Moderate Resolution Imaging Spectroradiometer (MODIS) images, together with in situ sampling for Southern California coastal waters, is shown in [23]. Holt et al. [15] performed a study on stormwater runoff plumes by comparing SAR and MODIS Aqua ocean color images, together with in situ measurements. They suggested a concept to complement water quality monitoring based on SAR and optical sensor data to guide the in situ collection and assessment of beach closures due to contamination. Devlin et al. [24] conducted an extensive overview of water quality and river plume monitoring in the Great Barrier Reef based on ocean color satellite data. The article provides an elaborate assessment of different water quality products based on remote sensing imagery.

The Black Sea is a semi-enclosed, brackish basin with a positive water balance [25], which makes river plume monitoring and mapping an important topic. Recent research by Lebedev et al. [26] used the concentrations of suspended matter acquired from a Medium-Resolution Imaging Spectrometer (MERIS) on board Envisat to establish a relationship between river runoff and the river plume area for two rivers on the Russian coast in the northeastern Black Sea. Osadchiev and Sedakov [27] applied an optical flow algorithm on near-simultaneous ocean color satellite imagery from Landsat 8 and Sentinel 2 to study the

spread of small river plumes on the northeastern shore of the Black Sea and successfully reconstructed surface currents along the shore. This method is efficient for detecting motion of frontal zones associated with river plumes that are visible in optical satellite data. Another study by Kostianoy et al. [28] utilized imagery from the MERIS, Operational Land Imager (OLI) and Thermal Infrared Sensor (TIRS) onboard Landsat 8, and the MultiSpectral Instrument (MSI) on Sentinel 2, to derive the total suspended matter (TSM) in order to investigate the behavior of stormwater river plumes along the Black Sea Turkish coast.

The remote sensing of wastewater plumes is a sparsely studied topic in the literature. There are several published papers using Landsat, Aqua, Terra and SeaWiFS data for the area of the Southern Californian coastline, USA. There is more research on stormwater runoff and river plume monitoring and mapping utilizing both optical and SAR data. However, recent publications hardly make use of the current Sentinel missions and do not benefit from their good temporal and spatial resolution.

This highlights the importance of investigating various water basins, to better understand the particularities of the processes and benefit from the improved image quality and resolution of current missions. An additional difficulty is the variety of optical properties in the individual basins and sometimes even in different areas of the same basin.

The Black Sea water is classified as Case 2, which makes its ocean color remote sensing a complex and challenging task [29]. Case 1 waters are those with IOPs dominated by phytoplankton (e.g., most open ocean waters) and Case 2 water basins (e.g., some coastal and inland water basins) are characterized by relatively high CDOM absorption and concentrations of non-pigmented particulate matter, which do not co-vary in a predictable manner with the chlorophyll-a (chl-a) concentration [30–32]. The standard bio-optical algorithms developed for Case 1 waters fail to describe the optical complexity of the Black Sea, giving large uncertainties (of the order of hundreds of percent) for chlorophyll-a concentrations in the coastal areas [33,34].

The present study introduces a distinct and innovative approach for the detection of effluent water, by using the fractions of satellite-image-derived absorption coefficients of phytoplankton pigment, CDOM and detritus instead of the concentration values, avoiding dealing with the different value ranges of the variables in areas with different bio-optical characteristics. In this way, we avoid the uncertainty arising from the algorithms for the calculation of concentrations and decrease the dependence on the ambient water characteristics, such as temperature and salinity.

The analysis is a first step towards the detection of effluent water and provides a framework for an automatic procedure to achieve this. We investigate the sea water properties near the Bulgarian coast and attempt to find IOPs that characterize different types of water—for example, riverine, coastal or open-sea water. Thus, an anomaly value that deviates from the typical IOPs on the satellite image could be a sign of the presence of an unusual water source in the area.

## 2. Methods

### 2.1. Construction of the Study Domain

The main challenges for detecting wastewater inflow relate to the small scale of the observable events and the strong influence of meteorological conditions on the surface signature. Additionally, the interaction of the effluent with the ambient marine water depends strongly on the wastewater constitution and thus makes a general description of the process difficult. Furthermore, the analysis requires substantial computational resources if the raw data are processed. In our study, these challenges are approached by studying the riverine plume surface signatures, which has a similar manifestation in satellite data, and comparing them with areas located in the discharge zone of wastewater treatment plants in order to identify common traits in the detection techniques.

Both inflows, from river and untreated wastewater pipes, are expected to appear similarly, as they have higher nutrient concentrations compared to the ambient sea shelf waters. Riverine runoff waters are the main nutrient sources in the sea, as they carry the

surface runoff from agricultural fields and farms, vehicle emissions and treated wastewater discharges. Similarly, characteristic of untreated wastewater are increased nitrogen and phosphorus concentrations, caused by human waste, food or certain detergent types. These nutrient carriers could also cause the appearance of organic matter not typical of the marine environment from both land sources. The high sediment concentration and the low salinity, compared to sea water, in both land sources is another common characteristic. On the other hand, treated wastewater effluent is fresh, compared to the ambient sea water, and should be low in nutrients and sediment concentration.

In the current study, we use optical remote sensing data and calculate the Inherent Optical Properties (IOPs) for our region of interest. Analyzing the IOPs is a logical choice as they are the primary output of the C2RCC processor and thus less prone to errors due to parameter uncertainties in the computing algorithm [35].

To be able to track differences between both types of land sources and see how they change with increasing distance from the shoreline and sea depth, we study their IOPs and compare them to inwater study points. The latter are located at the sea surface, away from the shoreline, at varying sea depths. Comparing these areas of interest offers the possibility to understand the characteristics of each and see how they vary depending on sea depth and land origin.

The study area of this research covers the Bulgarian Black Sea coastal to open-sea region. The IOPs are calculated for circular polygons, centered at river estuaries, in areas of wastewater effluent discharge and on the sea surface of inwater points, chosen at different sea depths and distances from the shoreline (Figure 1). The surface of the polygons, connected to land sources, is approx. 30,000 m$^2$ and the surface of inwater polygons is approx. 170,000 m$^2$. The distance between the polygons is of the order of 20 km.

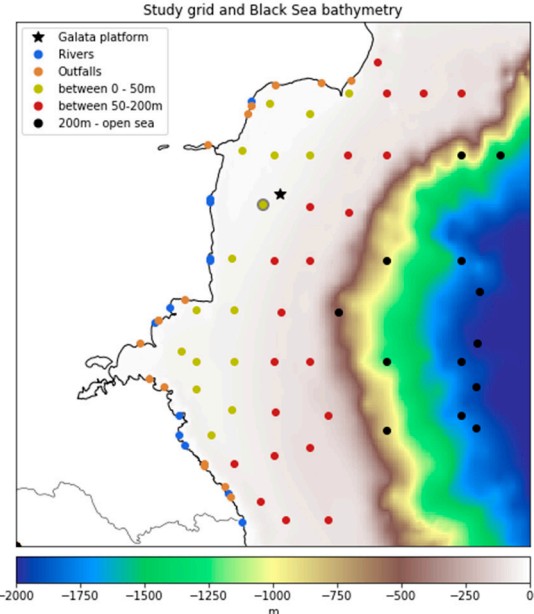

**Figure 1.** Bathymetry of the study area with grouping of the sites according to their source type and depth. Each point represents the area of ~30,000 m$^2$ for the river and outfall points and ~170,000 m$^2$ for the inwater points. The black star marks the coordinates of the Galata AERONET-OC site.

The choice of the coordinates and distribution of the study points was made as follows. For the land-sourced group, the positions of the estuaries of the main rivers flowing into the Black Sea, as well as the exact locations of the wastewater treatment plants, located on the sea coast, were determined. The points associated with river sources were located approximately 200 m away from the shoreline, in order to ensure that all analyzed pixels were water pixels and that uncertainties in the calculations, caused by bottom reflections due to overly shallow waters, were avoided. The points associated with wastewater

effluents were located approximately 100–150 m away from the shoreline, in the area where the wastewater treatment plant discharged the treated effluent.

The inwater points were located on the sea surface, away from the shoreline, and were grouped according to their corresponding sea depth. The aim was to see how the bathymetry (e.g., the change in the water circulation) as well as the distance from the shore influenced the variation in the IOPs. The three inwater depth groups were 0–50 m (excluding the land-sourced points); 50–200 m and from 200 m to open-sea areas with depths up to 2000 m. The locations of the study points are shown together with the bathymetry of the region in Figure 1.

For IOP calculation at a certain point, it is important to determine the value that is representative for the particular area. This is achieved by retrieving the IOPs for the number of pixels around the points of interest and taking the median value. For this purpose, a circular polygon around each point of interest was constructed and the IOPs retrieved by the C2RCC processor were calculated for each pixel inside the polygon. For the points associated with land sources, the radius of the circular polygon was approx. 100 m, and for the inwater points, it was approx. 230 m. The points associated with land sources had a smaller polygon surface compared to the inwater points, as the processes that we were interested in close to the shore were of small scale and could strongly vary away from the observed source. On the contrary, for the inwater points, it was more important to determine the general range of the values, which is not influenced by small-scale local phenomena.

Altogether, 84 positions were pinned, distributed on an even grid on the sea surface. The positions of interest were grouped into five categories. The first two groups represented all polygons in relation to land sources (blue and orange dots in Figure 1). These included:

- Group 1—12 polygons at river estuaries;
- Group 2—15 polygons in direct proximity to known wastewater treatment plants.

The remaining three groups contained the 57 inwater polygons, positioned at the sea surface (yellow, red and black dots in Figure 1). These polygons were grouped according to the bottom depth; thus, the coastal water/river runoff influence would weaken moving away from the shore. The surface circulation of the open Black Sea is characterized by a strong rim current [36], following the narrow continental slope, which limits the exchange of water and material between the coastal zone and the open Black Sea.

The three inwater groups that we defined were as follows:

- Group 3—17 polygons of sea depth up to 50 m; yellow dots in Figure 1;
- Group 4—24 polygons of sea depth between 50 and 200 m; red dots in Figure 1;
- Group 5—16 polygons of sea depth between 200 and 2000 m (open-sea region); black dots in Figure 1.

The bathymetry data were taken from the General Bathymetric Chart of the Oceans (GEBCO) [37] with a resolution of 30 arcsec.

In order to obtain a representative long time series, the IOP data were calculated for each polygon pixel of the study sites, for all acquisition dates available. Unfortunately, due to local cloud coverage, not all areas had the same representation throughout the year.

## 2.2. Satellite Data and IOP Retrieval

The satellite images that we found most reliable for our purposes were acquisitions of the Sentinel 2 MSI as it provides good spatial resolution of 10 m and a 5-day revisit time in the European mid-latitudes.

The IOPs were calculated with the Case 2 Regional CoastColour (C2RCC) processor included in the Sentinel Applications Platform (SNAP) for complex Case 2 waters [35]. All cloudless Sentinel 2 images acquired for the area of interest were processed for three years from 2017 until 2019. The C2RCC processor is a neuron network algorithm trained on a large, simulated dataset of water leaving reflectance and top-of-atmosphere radiances, retrieving the IOPs for optically complex coastal waters. The calculated IOPs were given

by C2RCC only for the 443 nm wavelength. Based on the IOP retrieval, the concentrations of the three optically active components—chlorophyll-a phytoplankton pigment, inorganic suspended sediments and yellow substances—were obtained. These were calculated from the IOPs by using locally adapted scaling factors.

C2RCC calculates the IOPs using local processing parameters, which describe the sea state and atmosphere. In our study, the sea surface temperature and water salinity were adjusted according to the time period that was being processed. Local sea surface temperature (SST) means for the three years were calculated from the Copernicus Marine Services (CMEMS) reanalysis data product [38] and are shown in Table 1. The annual mean surface salinity was calculated as 17.6 psu. The atmospheric pressure was set to 1000 hPa and the total ozone to 300 DU. In addition, several calculations for the sensitivity of the algorithm to the SST and salinity values were performed, and it was concluded that the seasonal average value was sufficient to ensure the stability of the estimates.

**Table 1.** Regional processing parameters for the SST in the C2RCC calculations.

|        | SST (°C) |
| --- | --- |
| Winter | 7.2 |
| Spring | 15.6 |
| Summer | 23.5 |
| Autumn | 14.6 |

The input of the C2RCC processor was the top-of-atmosphere reflectance from the Sentinel 2 MSI Level 1C products and the output were the five IOPs: three components for the total absorption—phytoplankton pigment absorption ($a_{pig}$), detritus ($a_{det}$) and CDOM ($a_{CDOM}$)—and two for the scattering—a white scatterer ($b_{wit}$) and a typical sediment scatterer ($b_{part}$). The total absorption at 443 nm $a_{tot}$ was given by the sum of phytoplankton pigment, detritus and CDOM absorptions $a_{tot} = a_{CDOM} + a_{det} + a_{pig}$ and the total scattering at 443 nm was the sum of the white scatterer and typical sediment scatterer $b_{tot} = b_{wit} + b_{part}$.

### 2.3. Variables Used in the Analysis

The variables analyzed and presented in this study were derived from the three above-mentioned absorption coefficients: $a_{pig}$, $a_{det}$, $a_{CDOM}$. Analysis of the absolute values of these coefficients is not optimal when trying to find a consistent pattern to characterize the optical properties of the water. We found it more appropriate to use instead the fraction of each of the three variables to the total sum $\frac{a_{CDOM}}{a_{tot}}$, $\frac{a_{det}}{a_{tot}}$, $\frac{a_{pig}}{a_{tot}}$ and analyze their percent contribution to the total absorption. This allows the identification of the predominant absorber for a local area and its variation with time. This also enables the comparison of processes specific to the coastal and deep-water regions.

Another benefit of analyzing the fraction contribution of the IOP absorption coefficients to the total is that we can reduce the impact of uncertainty caused by the empirical algorithm. Working with satellite-retrieved concentrations for detecting pollution in unknown water constitutions needs an in situ dataset to validate the results and support the conclusions.

We tackle the challenge of dealing with uncertainties caused by the complexity of the Black Sea waters by taking the primary output of the algorithm and additionally calculating the fraction of each value to the total, reducing the impact of retrieval uncertainties. One more advantage of this approach is that we reduce the necessity of information for the physical variables of the water.

### 2.4. Comparison with Previous Research

Previous research on the topic of wastewater detection, river plume and stormwater monitoring uses data for chlorophyll concentrations, turbidity, TSM and CDOM absorption, retrieved from algorithms for optical satellite data [10,11,16,26,28]. These studies base their

analysis on concentration anomalies and use additional in situ data for validation. Some authors track the normalized water-leaving radiance or remote sensing reflectance (Rrs), avoiding working with concentration values [13,14,39]. Stormwater and riverine inflow are fresh compared to the marine waters and often of lower temperature, so that changes in SST and salinity are indicative of these water masses [10,11,28].

Flattening of the sea surface roughness, observed on SAR images, can also be used for the visualization of stormwater runoff and wastewater diversion events [10,12,15]. These datasets are used complementary to detections on optical imagery, as sea surface roughness can be influenced by various other factors and is not sufficient for definite release detection. Holt et al. [15] tackle this problem by observing not only the common C-band SAR but also L-band from the Phased Array type L-band Synthetic Aperture Radar (PULSAR) instrument onboard the Advanced Land Observing Satellite (ALOS-1).

In our research, we concentrated on the optical Sentinel images in attempt to distinguish different water types by determining their predominant absorber, based on IOP fraction comparison. This is a new methodology, which is beneficial for studying a region such as the Western Black Sea, where common optical algorithms fail and in situ data are scarce.

In our preliminary analysis, we found that the SST and sea surface salinity (SSS) anomalies were not a reliable indicator for our study region. First, the wastewater discharges occurred in direct proximity to the shore; thus, the surface signature was of rather small scale. Second, the salinity of the ambient water was approximately 17.6 psu (see Section 2.2) and the mixing in the coastal region caused the fresh plumes to disappear quickly. As for the SST, it is mainly dependent on the meteorological conditions, and strong winds often cause up- and downwelling events, which could complicate the interpretation of the results for the effluent plumes.

## 3. Comparison of the C2RCC Results with In Situ Data

The C2RCC processor computes the chlorophyll-a concentration directly from the phytoplankton pigment absorption using the equation $Chl_a = 21 * a_{pig}^{1.04}$ [40].

The chlorophyll data calculated with C2RCC were compared with the data collected at the Galata AERONET-OC site [41], located ~23 km from the coastline southeast of Varna. The position of the platform is marked with a black star in Figure 1. The closest study point from our grid was located at ~6 km distance from it and is marked with a grey circle in Figure 1. The AERONET data were collected and processed with standardized instruments and calibration procedures in near-real time, with the primary output of the water-leaving radiance $L_w$ at different center wavelengths $\lambda$ in the 412–1020 nm spectral region and the number of secondary variables, among which is also the chlorophyll-a concentration. The analyses were based on fully quality-controlled AERONET-OC Level 2.0 products [42].

Figure 2 shows both datasets for the studied time period. The visual inspection indicates a good correlation between the two curves. Unfortunately, not all data coincided in time; thus, we calculated the Pearson correlation coefficient for values measured on the same day. It was 0.73, which indicates a good correlation. A major inconsistency between both datasets was observed in the beginning of the summer months in 2017, from the beginning of May until the beginning of August. A possible reason for the discrepancy is the reported intensive phytoplankton bloom of *Emiliania huxleyi* [43]. The chlorophyll-a concentration from the Galata platform was derived from an algorithm, which could give errors in unusual circumstances.

*Emiliania huxleyi* is photosynthetic plankton with calcite covering, causing increased backscattering, which is not well detectable at absorption scale [44]. The spectrum of the above water remote sensing reflectance of *E. huxleyi* coccoliths is characterized by a maximum in the wavelength range of 450–500 nm depending on the size and morphology of single coccoliths [45]. An extensive study of the remote sensing reflectance from multispectral satellite ocean color instruments in different optical water types demonstrated a characteristic peak at 490 nm for *E. huxleyi* blooms in various water basins [46]. The

remote sensing reflectance is the ratio between the water-leaving radiance $L_w$, which is the upwelling radiance from the sea surface to the air, and the downward irradiance above the sea surface $E_d$, and thus it is proportional to $L_w$ [47].

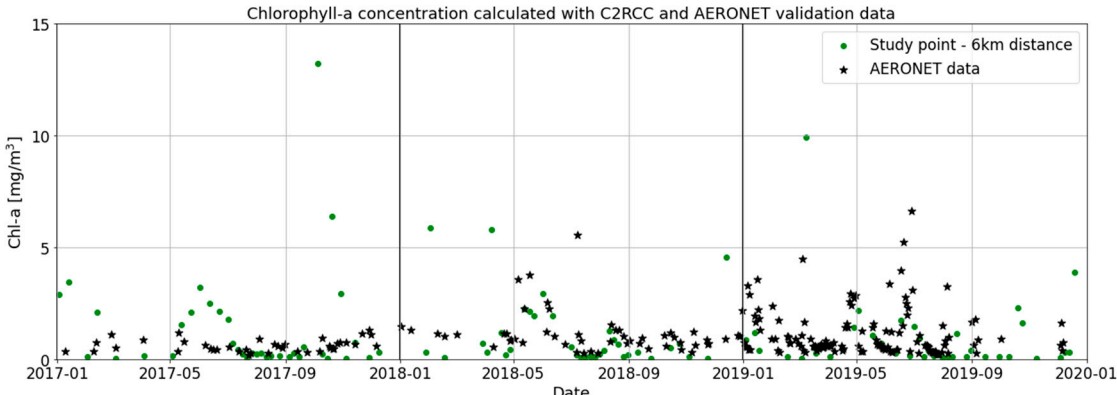

**Figure 2.** Chlorophyll-a concentrations in (mg/m³) in the period 2017–2019. Black stars represent the data from the AERONET-OC Galata site and the green dots are the concentrations calculated with the C2RCC processor for the nearest study point.

In situ AERONET-OC data of the water-leaving irradiance $L_w$ for different wavelengths $\lambda$ are presented in Figure 3, and a maximum for $\lambda$ = 412, 443, 490, 532 and 551 nm is clearly distinguishable for the above-mentioned summer period in 2017. A slight increase is also observable for the summer months of 2018, with very low intensity. During the period 23 May–14 June 2019, a bloom event was observed, with lower intensity and duration compared to those in 2017 [43], and it is also visible in Figure 3.

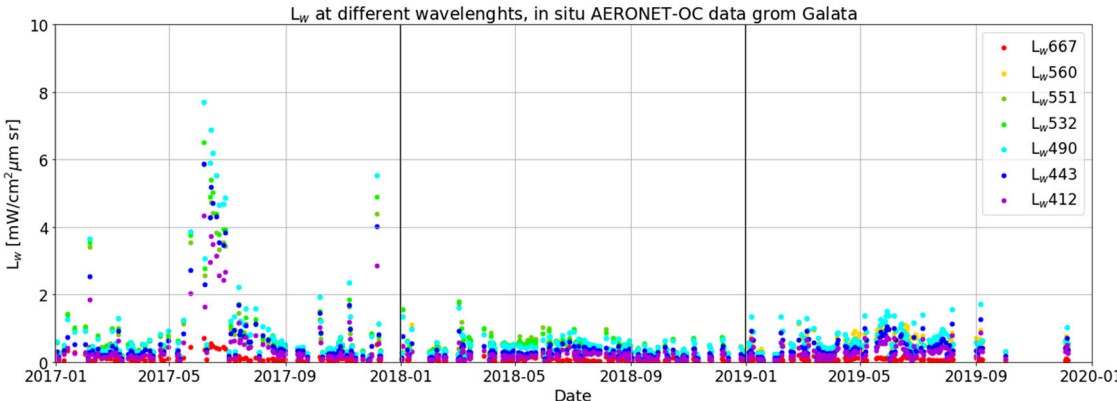

**Figure 3.** Water-leaving radiance measured for different wavelengths at Galata AERONET-OC site in the Black Sea.

The values of the water-leaving radiance $L_w$ at 443 nm measured at the Galata platform were compared to the chlorophyll-a concentration and the backscattering of marine particles at 443 nm, calculated by the C2RCC processor at the verification study point. They are shown in Figure 4, which illustrates the water-leaving radiance at 443 nm at the Galata AERONET-OC site, the chlorophyll-a concentration and the scattering of marine particles at 443 nm, calculated with C2RCC.9.

The graphs of the chlorophyll-a concentration (green dots) and the water-leaving radiance (black stars) show generally similar progression, although the values of chlorophyll-a reveal greater variability between various acquisition days and have several outlier values. The peak during the bloom period visible in the backscattering graph (yellow dots) confirms that the event could have been caused by *E. huxleyi* as they reflect the incoming light strongly.

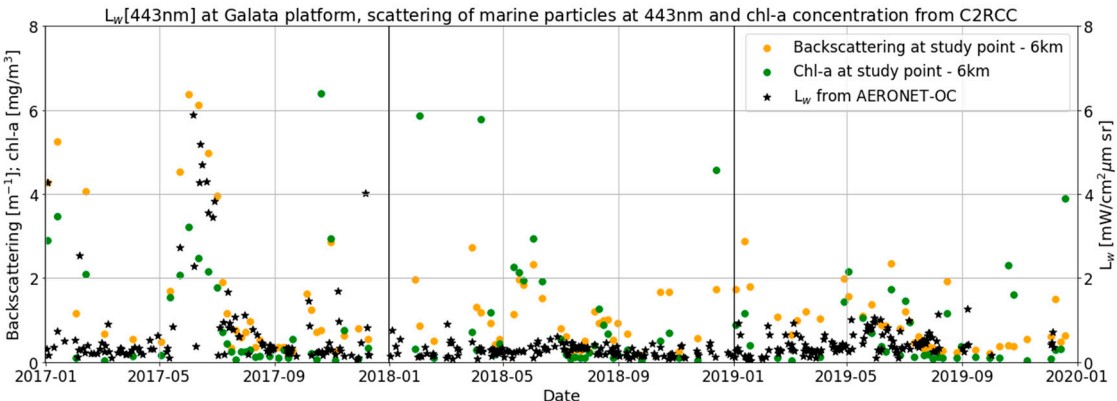

**Figure 4.** Water-leaving radiance at 443 nm at Galata AERONET-OC site (black stars), chlorophyll-a concentration (green dots) and scattering of marine particles at 443 nm (yellow dots) calculated with C2RCC.

The main feature of the three graphs is that they exhibit the distinctive peak for the time of the bloom in 2017 and reveal very good overlap regarding the timespan of the event and the intensity. A major drawback of the values calculated by C2RCC is the variability between single measurements with significant outlier points. This can be attributed to distortions due to partially cloudy polygons and uncertainties due to the particularities of the ocean color observations in the Black Sea region.

The comparison of the C2RCC calculations with the measurements from the Galata platform confirms the validity of C2RCC for the observed region of interest. In general, the processor correctly reveals the temporal progression of the computed variables; however, for some individual acquisitions, it gives inconsistently high values.

These observations stress the importance of the local adaptation of ocean color algorithms for chlorophyll-a retrieval for the Black Sea region for both the C2RCC processor and the algorithm used at the Galata validation platform.

## 4. Analysis of the IOP Fractions

The percent contribution of the three IOP fractions to the total absorption were calculated for each of the 84 polygons of interest for all cloudless acquisitions in the period 2017–2019. The median value of each polygon was calculated, and results were sorted according to the five groups (two land-sourced groups and three groups of inwater points) described in Section 2.1.

### 4.1. Daily Time Series of the IOP Values in 2017–2019

The timeline graphs presented in Figure 5 summarize the results for the two main types of study polygons: the land-sourced and those away from the shoreline. The land-sourced study area includes the data from all points related to land sources, such as rivers (Group 1) and outfalls (Group 2). The other study area covers the area away from the shoreline up to open sea and it includes all inwater points (Groups 3, 4 and 5). For each of the two areas, the median value of the points of interest was taken for each day with cloudless acquisition. Taking the median avoids the influence of dominant local events such as downpours or intensive agricultural activities, and outliers are sorted out. This gives more representative information about the particularities of each of the two groups [48].

The IOP fraction time series from the land-sourced groups is presented in Figure 5A. The predominant absorber is CDOM, contributing to over 50% of the total absorption for the time from late March till the middle/end of December for all years. Detritus and phytoplankton share a similar portion of the remaining absorption, varying by around 20–30%. The change in the percent contribution in detritus or phytoplankton pigment is compensated between them, as CDOM retains its high values.

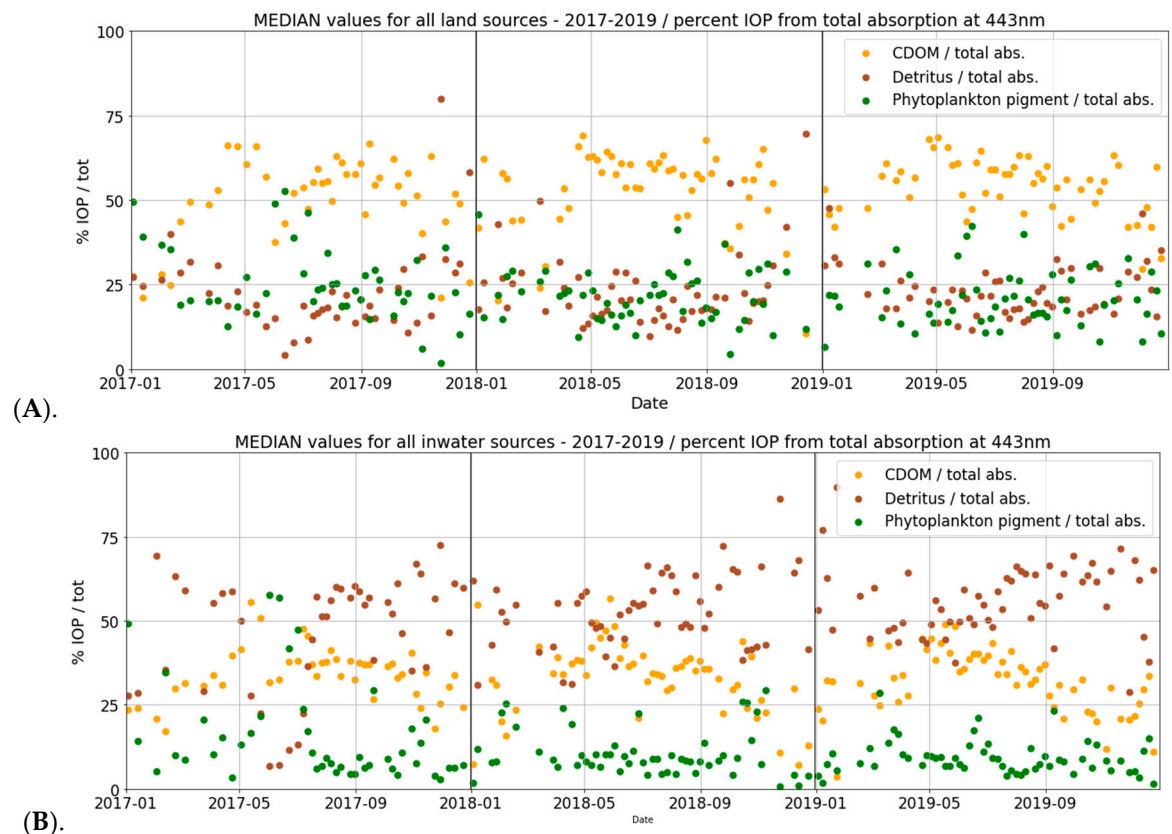

**Figure 5.** Time series of the percent contribution of each IOP fraction to the total absorption at 443 nm (CDOM—yellow, detritus—brown, phytoplankton pigment—green); (**A**). The median values of all fluvial points and those connected to wastewater outfall (groups 1 and 2); (**B**). The median value of all inwater study points, for all sea depths (groups 3, 4 and 5).

The observed temporal fluctuations are small. For all three years, the winter period from late December until the beginning of March is marked by a gradual decrease in CDOM absorption to 25–30% and an increase of detritus and phytoplankton pigment absorption to above 30%. An intensive phytoplankton bloom was detected in the beginning of the summer season in 2017, with a spike in phytoplankton pigment, a minimum value for detritus and a small reduction in CDOM absorptions. Such an event, but with lower intensity, could be observed also in 2019. The year 2018 did not show clear indications for a bloom event.

The time series of the inwater groups are shown in Figure 5B. The predominant absorber for inwater points is detritus, contributing to approximately 50–60% of the total absorption at 443 nm, followed by CDOM with 30–40%. The phytoplankton pigment absorption is the lowest, with less than 20%.

This behavior changes during the period of phytoplankton bloom, as in the land-sourced groups. During the early summer period, there was an observable peak in the phytoplankton pigment fraction contribution, which was compensated mainly by a decrease in the detritus fraction contribution. The CDOM contribution shows a small decrease. The bloom was recorded with the strongest intensity in 2017, and another bloom was visible in 2019 with considerably lower intensity. There were fluctuations in the fraction of the phytoplankton pigment absorption in the spring/early summer of 2018; however, they cannot be clearly attributed to a bloom event.

### 4.2. Ternary Graphs—Annual Means

More characteristics of the absorption coefficients of the different groups become visible when presented as ternary graphs. The ternary graph shows the fraction contribution of CDOM, detritus and phytoplankton pigment to the total absorption budget in percent

on three axes. On the ternary plots, temporal variation is not visible; however, peculiarities related to each group become more prominent.

Figure 6 illustrates the ternary graphs for all five groups. Each group is presented in a different color. On the first line are the two land-sourced groups and on the second are the three inwater groups, sorted according to the sea bottom depth. Each single point on the graph represents a measurement from an available acquisition for one of the study polygons within its group. The graph allows the direct comparison of all five study groups according to their absorption range and the identification of the differences between them.

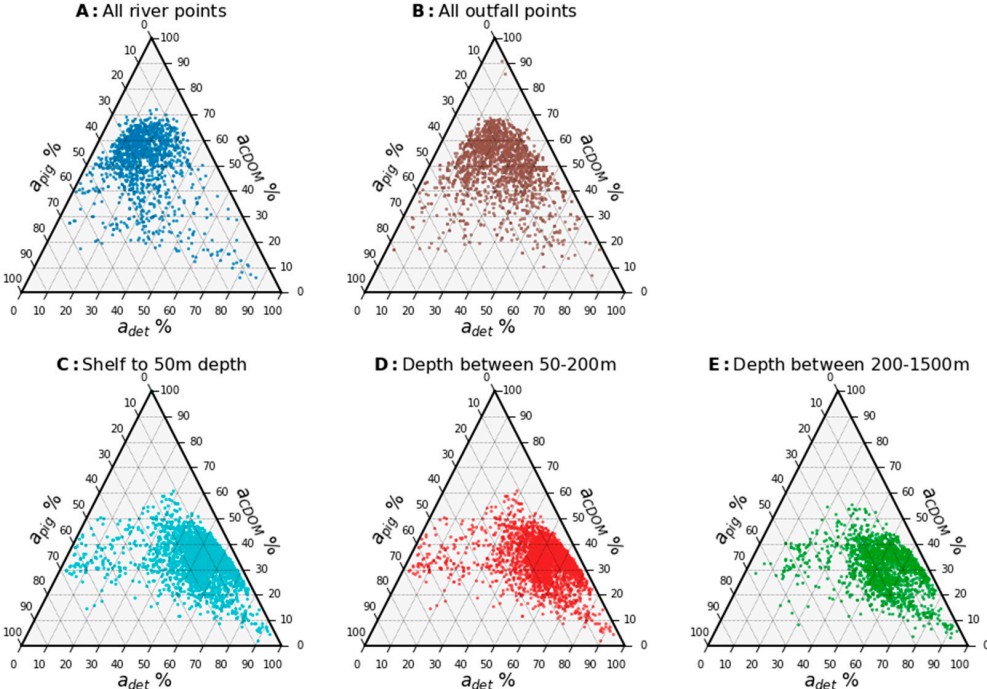

**Figure 6.** Ternary graphs for all five groups (land-sourced—upper line (**A**,**B**), inwater points—second line (**C**–**E**)). All points for all acquisitions are plotted for each group.

The two land-sourced groups, plotted in Figure 6A,B, reveal very similar behavior, with the majority of values situated in the area of increased CDOM and equal detritus and phytoplankton pigment shares. In particular, in contrast to the outfall points, the river sources reveal several points in the lower-right region, associated with very high detritus and very low CDOM and phytoplankton pigment levels. Additionally, the river group shows a cluster of points concentrated around the lower center with equally distributed CDOM, detritus and phytoplankton pigment fractions. The outfall group has these points equally distributed in areas with extreme high or low detritus fractions.

The three inwater groups are presented in Figure 6C–E. The first two groups of study points—sea depth at the shelf up to 50 m depth and from 50 up to 200 m—show points in the middle-left area that are lower in the last group. They represent very low detritus levels, high phytoplankton pigment levels at around 70% and a moderate CDOM contribution. This percent contribution can be attributed to an intense phytoplankton bloom, which is also visible for the land-sourced groups. Similar to the river group, all three inwater groups reveal some points with very high detritus and negligibly small CDOM and phytoplankton pigment fractions.

An important difference between the three inwater groups is that the deep-water study group shows most measurements with a CDOM percent contribution lower than 40–45%. The other two inwater groups have CDOM reaching up to 50%.

Figure 7 represents the seasonally and annually averaged values from the graph shown in Figure 6. The five study groups are color-coded and the years are presented in different color intensities. The months of averaging were chosen in accordance with

the slower warming and cooling of water compared to the atmosphere: winter—Jan, Feb, March; spring—April, May, June; summer—July, August, Sep; autumn—Oct, Nov, Dec.

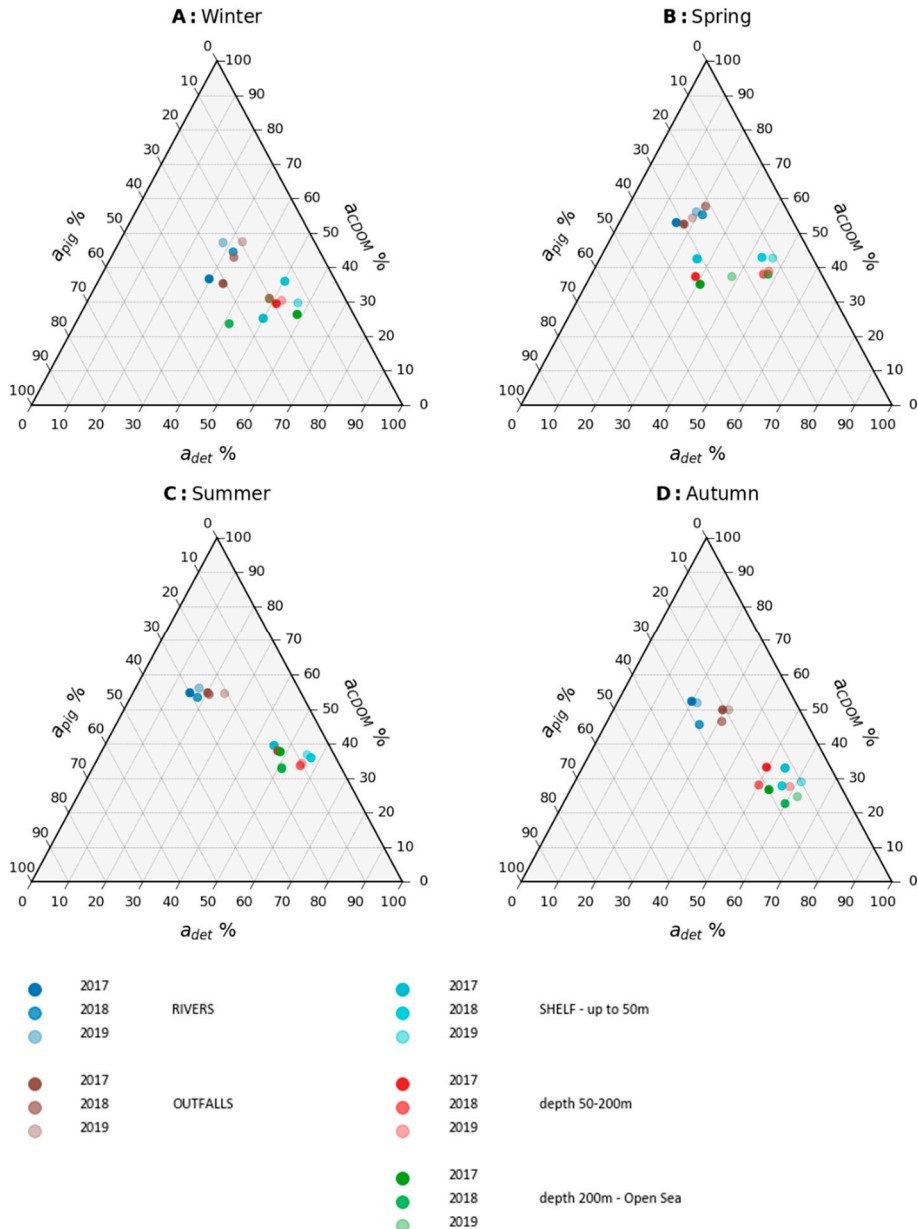

**Figure 7.** Ternary diagrams of the seasonal and annual means for all five groups. The percent contribution of each of the three IOPs to the total absorption at 443 nm is plotted on the three axes. The mean value of each group is plotted for each of the years 2017–2019. The legend below gives the color code for each group and year.

On this ternary diagram, the study points associated with land sources are well distinguishable from the inwater points for all seasons and years. Their increased CDOM absorption and decreased detritus fraction contribution distinguish them from the inwater points.

For the cold part of the year (winter and spring), the inwater groups show greater variability for all three fraction contributions. The phytoplankton bloom visible in Figure 2 for 2017 is also recognizable on this graph. The bloom peak occurred in June (Figure 7B: spring), which is distinguishable by the increased phytoplankton pigment and decreased detritus contribution. For this year, the gradual decrease in CDOM away from the shoreline is visible as well. The warm part of the year (summer and autumn) shows a greater

difference between the land-sourced and the inwater groups and less variability between individual groups.

Figure 8 is a ternary diagram, which gives the time-averaged values of the 84 polygons for all three years. The points are color-coded according to their group (same colors as in Figure 7). This figure clearly illustrates the differences among the five groups.

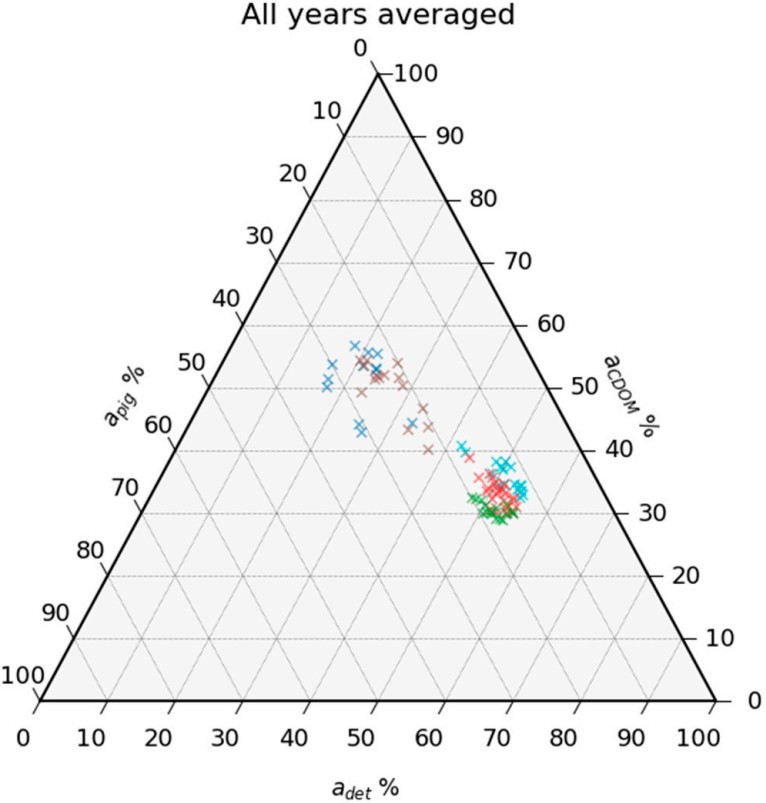

**Figure 8.** Ternary diagram of the averaged values for all three years for each point of the five study groups. The five groups are plotted in the same colors as in Figure 7.

The points from the two land-sourced groups are well-separated from the three inwater groups according to their enhanced CDOM percent contribution.

The land-sourced groups 1 and 2 give two separated clusters, with some points within them overlapping. The separating values are mainly CDOM and detritus contributions to the total absorption.

In Figure 8, the separation between the three inwater groups, 3, 4 and 5, is clearly visible. They are separated according to their phytoplankton pigment fraction contribution, which increases with increasing sea bottom depth. Another fraction that depends on the sea depth is the CDOM contribution. It decreases with increasing sea depth, albeit at a slower pace compared to the phytoplankton pigment fraction.

## 5. Framework for an Automated Procedure for Water Type Classification

The analysis described above allows us to draw a framework for an automated procedure to distinguish between land-sourced and inwater groups of points. Figure 9 presents the workflow diagram individuating the necessary steps required for classification. The procedure could be repeated in several iterations, in order to reach smaller spatial scales and better identify the signal of the effluent water.

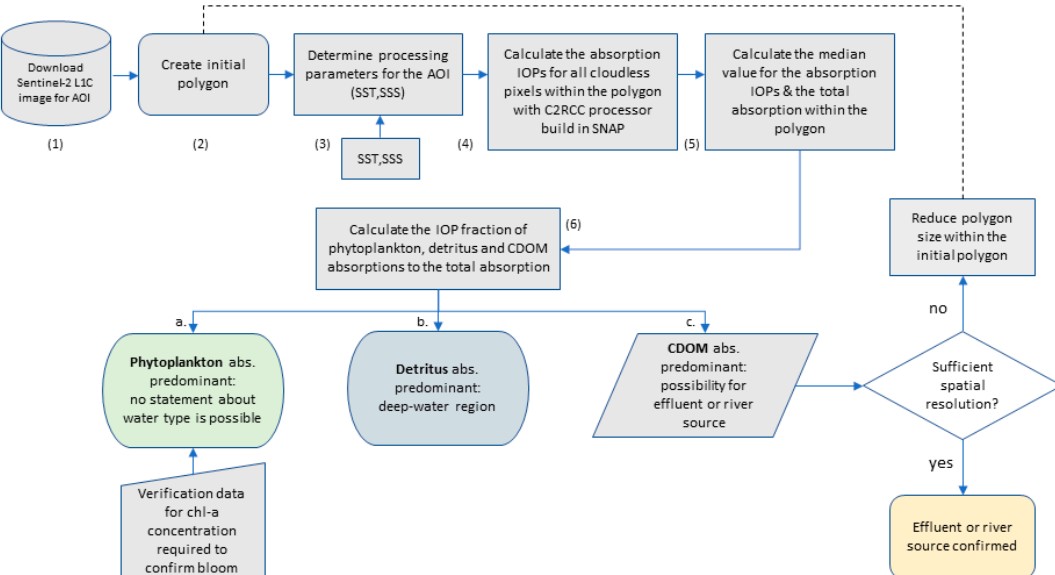

**Figure 9.** Flowchart describing the workflow for water type classification.

First, the user is required to download the Sentinel 2 Level 1C image for the area of interest (AOI) (1) and to create a polygon around the area of suspected wastewater release (2). The processing parameters for SST and salinity (3) are necessary input for the calculations with the C2RCC processor (4). The output of the processor are three IOPs for the absorption coefficients—for phytoplankton pigment, CDOM and detritus—and the total absorption, i.e., their sum, for each pixel within the polygon. Then, the user should calculate the median value for the IOPs for the pixels within the polygon (5). For the three absorption IOPs, the fraction to the total absorption is calculated (6). Comparing the fraction contribution, three cases can be distinguished:

a. Phytoplankton pigment fraction is predominant: There is the possibility of phytoplankton bloom and no statement about the water type can be made. Confirming the presence of the bloom can be achieved with in situ sampling or via other information sources.
b. Detritus fraction is predominant: The polygon is located away from the shoreline and no influence of riverine or effluent plume is present.
c. CDOM fraction is predominant: There is a possible influence of wastewater or a river plume. If the initial polygon size is too large and the user requires more precise tracking, the procedure could be repeated with smaller sub- or neighboring polygons, returning to step (2). Thus, a more precise signal of the effluent water could be obtained with a better spatial location.

## 6. Discussion

One of the main findings in our analysis is the predominant CDOM absorber in the land-sourced water. Typical for fluvial waters is their increased turbidity and nutrient concentration. Particulate matter released from the riverbed and through soil erosion makes river waters more turbid, and, when draining into marine environment, their plume appears with a specific brown, yellowish color on true color images. The particulate matter contains optically active absorbers, which lead to the increased CDOM absorption fraction in the coastal region. This mechanism is a possible explanation of our results showing increased CDOM values for all land-sourced points.

Other authors also find that the CDOM is a major feature that characterizes the fluvial water. Marmorino et al. [17] identified wastewater discharge off the southeast coast of Florida, USA, through its elevated CDOM levels. Studies based on absorption spectroscopy measurements have shown similar results for the eastern Caribbean Orinoco River outflow [49] and for the Middle Atlantic Bight [50]. Campanelli et al. [51] demonstrated that

River Po in the North-Central Western Adriatic Sea is the main source of CDOM and nutrient circulation for the region, based on field measurements and MODIS Aqua data.

The late autumn and spring time is the period of most intense precipitation in the Bulgarian Black Sea region, leading to increased river inflow. This explains the small peak in the CDOM fraction contribution for all three years on the land-sourced graph (Figure 5A). The increased CDOM fraction is barely observed for the inwater polygons, as the impact of fluvial inflow decreases rapidly away from the shoreline.

The study polygons located away from the coast (groups 3, 4 and 5) indicate the highest absorption fraction due to the detritus, followed by CDOM. The phytoplankton pigment contribution has the lowest values, also lower than for the land-sourced points. This could be explained by the fact that, away from the shoreline, there are more dead particulate organic absorbers, by-products of zooplankton graze or dead phytoplankton cells, but this needs further investigation.

Churilova [52] investigated the phytoplankton and detritus absorption coefficients from 1 to 2 levels in the euphotic zone based on in situ sampling in spring 1995 in the central Black Sea. At the surface, the detritus contribution to the total varied between 23% and 62% at a wavelength of 440 nm. Efimova et al. [53] showed that, at a 438 nm wavelength, the non-algal particles' contribution to the total absorption was $40 \pm 15\%$ on the surface for the deep-water region and $54 \pm 11\%$ in the coastal areas. For CDOM, the contribution to the total was 69% on average, with almost unchangeable values in the water column. The in situ sampling was performed in spring 2019 for the central North Black Sea. This is in accordance with our results regarding the predominance of the CDOM component in the absorption fraction in the coastal part of the sea and detritus in the shelf part.

The phytoplankton bloom detected in the late spring to summer period in 2017 and spring 2019 was characterized by a peak in the phytoplankton pigment absorption fraction and a minimum value for detritus. This bloom was also observed by Cazzaniga et al. [43]. The bloom was observable with similar intensity also at the inwater study points, away from the shoreline, with similar variations in the phytoplankton and detritus absorption fraction. Consequently, the CDOM fraction's contribution to the total absorption in this period decreased slightly throughout the study domain.

During the summer period, the marine biological activity was, in general, decreased, and no significant variations in the three absorption fractions could be observed.

Ternary graphs are a good way to visualize and summarize the results, as the clusters of points give the typical range and interrelations between the three absorbers. The most important result from the clustering of the five groups of polygons is surely the separation of land-sourced from inwater polygons by the value of the CDOM fraction, which was seen in all seasons and years. This could be used as a basis to implement an algorithm to identify effluent and riverine water in the marine environment on satellite images. Moreover, another important result is that the absorber with the highest fraction for the inwater points is detritus, and, moving away from the shore, the absorption fraction of the phytoplankton increases.

The phytoplankton bloom, which occurred with high intensity in June 2017, is also distinguishable on the spring ternary plot (Figure 7B). It is identified by the increased phytoplankton pigment fraction contribution relative to the average during this period. The gradual CDOM decrease due to the weakening river impact away from the shoreline became visible as well.

## 7. Conclusions

Observing and detecting effluent plumes in the marine environment using satellite data is an especially challenging task, mainly because of the scale of the events, the particularity of the discharges and the variations in the marine conditions. In this study, we investigated the surface signature of riverine plumes, assuming that, in a similar way, we could apply the results to the treated wastewater as the two types of water share similar

optical characteristics. The study domain covered the Bulgarian Black Sea region from the shore to the open sea at a bottom depth of 2000 m, and it was separated into points at various distances from the shore and bottom depths. In this way, we were able to determine the extent of the riverine and outfall flows in the marine environment. The study points were grouped into five categories: riverine (near the river estuaries), outfall (near the known WW treatment plants) and inwater with bottom depth in the ranges 0–50 m, 50–200 m and more than 200 m. The first two groups were land-sourced and the other three were inwater.

The inherent optical properties of the different study points were calculated using the C2RCC processor for all available Sentinel 2 cloudless acquisitions for the years 2017, 2018 and 2019, and the fraction contribution of each of the three absorbers—detritus, CDOM and phytoplankton pigment—to the total absorption were calculated and analyzed.

Additionally, AERONET-OC optical data from the Galata platform were used to compare the chlorophyll-a concentration calculated by C2RCC for the nearest study point, located ~6 km away from the platform site.

In the paper, we analyzed the absorption fractions of the CDOM, detritus and phytoplankton pigment for the five groups of polygons in order to identify patterns to distinguish between various types of water. The main findings of this study can be summarized as follows:

- The validation data from AERONET for the chlorophyll-a concentration showed generally good consistency with the values calculated by C2RCC, proving the validity of the study methodology. The main mismatch was observed for the intensive phytoplankton bloom of *Emiliania Huxleyi* from May until August 2017. This peak was observable in the water-leaving radiance $L_w$ at 443 nm and 490 nm, with great overlap of the timespan and intensity of the event, stressing the importance of the regional fine-tuning of ocean color algorithms, used both for AERONET data processing and in the C2RCC processor, implemented in SNAP.

- The result of the IOP fraction calculation was that, for groups 1 and 2, close to the shore, the predominant absorber was CDOM, because river runoff carries a significant amount of optically active components. Changes in the detritus and phytoplankton pigment fraction were compensated between them. For inwater groups (3, 4 and 5), the predominant absorber was detritus. This result is in accordance with previous similar studies.

- We detected a phytoplankton bloom in the period May–August 2017 with a peak in the phytoplankton pigment fraction and a minimum detritus fraction. During 2018, we did not observe any clearly distinguishable bloom event. In late spring 2019, a bloom event with low intensity and short duration was visible in both datasets—the C2RCC calculated values and those from the AERONET-OC site.

- The impact of the fluvial plume decreased gradually away from the shore. The ternary plots clearly showed that the main characteristic of the fluvial and the outfall water was the increased value of the CDOM contribution to the absorption. The river and outfall groups were similar; however, the river group presented extreme values, with high detritus and low phytoplankton pigment and CDOM fraction values. In this study, the outfall group did not show signs of significant pollution load or extreme values.

- On the temporally averaged ternary plots, the surface signature of the three inwater groups could be clearly distinguished according to the predominant detritus contribution to the total absorption. The fraction of the phytoplankton pigment increased moving away from the shore towards the open sea, accompanied by a decreased CDOM fraction.

Based on these findings, we draw a framework that could be used for the development of an algorithm for the identification of unusual sources of water by a rapid change in the typical absorption fractions of the CDOM, detritus and phytoplankton pigment at the site. Analyzing the fraction contribution of the main absorbers instead of their concentrations is an innovative approach, which presents certain benefits: it decreases the uncertainty

arising from the algorithms for the calculation of concentrations (often misleading in areas with optically complex waters such as the Western Black Sea) and eliminates the need for detailed information on the physical properties of the marine water.

**Author Contributions:** Conceptualization, I.G. and E.P.; Methodology, I.G. and E.P.; Software, I.G.; Validation, I.G., V.S. and E.P.; Formal Analysis, I.G., V.S. and E.P.; Investigation, I.G. and E.P.; Resources, I.G. and V.S.; Data Curation, I.G.; Writing—Original Draft Preparation, I.G.; Writing—Review and Editing, E.P. and V.S.; Visualization, I.G.; Supervision, E.P. All authors have read and agreed to the published version of the manuscript.

**Funding:** This research received no external funding.

**Data Availability Statement:** The Sentinel-2 satellite images are publicly available on the Copernicus Open Access Hub https://scihub.copernicus.eu/ (accessed on 7 September 2021). AERONET-OC Level 2.0 data is publicly available on the Goddard Space Flight Center—AERONET Ocean Color website https://aeronet.gsfc.nasa.gov/new_web/ocean_color.html (accessed on 7 September 2021). Data for sea surface temperature and sea surface salinity are publicly available from the Copernicus Marine Services (CMEMS) website https://doi.org/10.25423/CMCC/BLKSEA_MULTIYEAR_PHY_007_004 (accessed on 7 September 2021).

**Acknowledgments:** The Sentinel 2 data were processed making use of the G-POD (Grid Processing on Demand) Service of the European Space Agency for Earth Observation Applications, with the kind support of Beniamino Abis and team. We wish to thank Giuseppe Zibordi, as PI of the Galata site, for the AERONET-OC Level 2.0 data product processing and site maintenance. The research was performed with open access data and software; satellite images and the SNAP architecture were made available by the European Space Agency, and AERONET-OC data were supported by NASA.

**Conflicts of Interest:** The authors declare no conflict of interest.

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
