# Peer review of "Detecting the Surface Signature of Riverine and Effluent Plumes along the Bulgarian Black Sea Coast Using Satellite Data"

_remotesensing, doi:10.3390/rs13204094_

Round 1
Reviewer 1 Report
I've just finish to read the ms. no. remotesensing-1392441, titled "Detecting the surface signature of riverine and effluent plumes along the Bulgarian Black Sea coast using satellite data" and I think it is worthy of publication but only after minor revisions. Specifically, Introdcution may be reduced without losing the text meaning and more info may be added about the difference between this study and previous ones. In this respect there are some contributions in literature (in the last 5 years) that ought to be cited. Additionally, the language seems to be scholastic in some sections; a help from a native-tongue colleague may be desirable.
Author Response
We would like to thank the anonymous reviewer for the dedicated time and the useful comments. Please find the replies to the remarks below.
'I've just finished to read the ms. no. remotesensing-1392441, titled "Detecting the surface signature of riverine and effluent plumes along the Bulgarian Black Sea coast using satellite data" and I think it is worthy of publication but only after minor revisions.
Specifically, Introduction may be reduced without losing the text meaning and more info may be added about the difference between this study and previous ones. '
The comment is taken into account. The Introduction is revised and a paragraph stating what is the main difference from other studies is added. Mainly, in our study we used the fractions of absorption due to detritus, CDOM and phytoplankton instead of the concentration. This way we avoid dealing with the different value ranges of the variables at different places. Furthermore, we decrease the uncertainty coming from different bio-optical algorithms to calculate the concentration values and the need to use information on the water physical characteristics (temperature, salinity etc).
'In this respect there are some contributions in literature (in the last 5 years) that ought to be cited.'
The comment is taken into account and we have reviewed again the published literature in the last years. The Introduction lists several more studies on the subject. We hope the added articles increase the list of relevant papers. However, studies on the optical properties in the Western part of the Black Sea near the Bulgarian coast using Sentinel missions are not so numerous.
'Additionally, the language seems to be scholastic in some sections; a help from a native-tongue colleague may be desirable.'
We take a note on this as non-English native speakers and intend to send the paper to the editing services offered by MDPI before the final submission.
Reviewer 2 Report
Summary
Effluent water is discharged directly into riverine or marine environment and in case of insufficient treatment it can have significant impact on the ecosystem, alter the state of biological communities or impact human health. Independent and continuous monitoring of such discharges is of great importance for the coastal zone management, sustainable development and blue growth of coastal regions. In this regard satellite re- mote sensing offers an irreplaceable information source. Therefore, the work of this paper is of great significance.
Observing and detecting effluent plumes in marine environment using satellite data is especially challenging task. This works analyses the absorption fractions of the CDOM, detritus and phytoplankton pigment for the 5 groups of polygons trying to find patterns to distinguish between various types of water. The structure is logical, the figures are of good quality and the historical background has given credit as is appropriate. The English and logic should be further improved substantially throughout the whole paper. There are too many “Error! Reference source not found”. Overall, I think this manuscript can be considered after major revision if the author could adequately address the comments below.
Specific/Detailed Comments
- A flow diagram to describe how to distinguish the different water types through the proportion of CDOM, detritus and phytoplankton pigment is necessary.
- Since that the author used the satellite image, but there are no satellite-based retrieved image results and classified image results for effluent water. Just a few discrete points could not reflect the advantages of remote sensing for large area. Nor does it fit the title of the article.
- It’s not a convincing using ternary graphs of the proportion of CDOM, detritus and phytoplankton pigment for distinguishing all five water groups. Effluent water signal is very weak and is very hard to be distinguished from ambident signal. And as the author mentioned, the salinity and temperature are key factors, why it was not used here.
- Comparisons with the others’ methods in the literature is necessary and convincing.
- Line 174. “Error! Reference source not found”.
- Line 195. “Error! Reference source not found”.
- Line 209. “Error! Reference source not found”.
- Lines 221,223,226,252…too many “Error! Reference source not found”.
Author Response
We would like to thank the anonymous reviewer for the dedicated time and the useful comments. Please find our replies to the remarks below.
'Summary
Effluent water is discharged directly into riverine or marine environment and in case of insufficient treatment it can have significant impact on the ecosystem, alter the state of biological communities or impact human health. Independent and continuous monitoring of such discharges is of great importance for the coastal zone management, sustainable development and blue growth of coastal regions. In this regard satellite re- mote sensing offers an irreplaceable information source. Therefore, the work of this paper is of great significance.
Observing and detecting effluent plumes in marine environment using satellite data is especially challenging task. This work analyses the absorption fractions of the CDOM, detritus and phytoplankton pigment for the 5 groups of polygons trying to find patterns to distinguish between various types of water. The structure is logical, the figures are of good quality and the historical background has given credit as is appropriate.
The English and logic should be further improved substantially throughout the whole paper. There are too many “Error! Reference source not found”. '
The comment is taken into account. We are sorry about the references' corruption, obviously something went wrong after the conversion of our file. It is corrected in the revised article. Regarding the language, we intend to send the paper to the editing services offered by MDPI before the final submission.
'Overall, I think this manuscript can be considered after major revision if the author could adequately address the comments below.
Specific/Detailed Comments
1. A flow diagram to describe how to distinguish the different water types through the proportion of CDOM, detritus and phytoplankton pigment is necessary.'
The comment is taken into account. Indeed a diagram shows better our idea how to identify effluent flows through consequently checking the fractions of CDOM, detritus and phytoplankton. A Section 5 on development of such an algorithm is written and the flow diagram is added as Figure 9. Furthermore, it is seen from the diagram, that the procedure could be iterative, and repeated second (or more) time for the polygons with increased CDOM contribution to the absorption, thus obtaining better spatial resolution, better signal and more precise location.
'2. Since that the author used the satellite image, but there are no satellite-based retrieved image results and classified image results for effluent water. Just a few discrete points could not reflect the advantages of remote sensing for large area. Nor does it fit the title of the article.'
The note is fair. The choice to work with “discrete” points was made in order to decrease the need of enormous computing resources when dealing with high resolution satellite images (~10 m resolution) for a 3 years period. Actually we took time on powerful virtual machines in order to process the original Sentinel data, but our resources were limited. However, the discrete points are an average value over a polygon of ~30000 m², thus each point represents an area and contains the information for all original pixels inside. After the first analysis of the polygons median representative value, one could repeat the same procedure for polygons with weak effluent signal and their neighbours after reducing the polygon size and centering it at areas of concern. After several iterations of this procedure one can better distinguish the signal of the effluent water in smaller space scales. This way we could work in smaller areas with less amount of data, doable on a good workstation. We have revised the text in Section 4 and added a Section 5 in order to better explain our idea and hope it is clearer now.
'3. It’s not a convincing using ternary graphs of the proportion of CDOM, detritus and phytoplankton pigment for distinguishing all five water groups. Effluent water signal is very weak and is very hard to be distinguished from ambident signal. And as the author mentioned, the salinity and temperature are key factors, why it was not used here.'
The comment is taken into account. The ternary graph was the natural tool to visualize the three components and it is complementary to the time series graphs from Figure 5, where one can get the same information extended in time axis. Looking at both graphs the separation between different groups is obvious. It is true that in some cases the points on the ternary graph overlap and it is difficult to distinguish the effluent water. To some extent the reason is that the points represent an average for the polygon (see the comment on point 2) and as the effluent water in this region is small-scale, the signal is mixed with the ambient pixels and weakens. One possible solution is to repeat the procedure for the smaller polygon and its neighbours and search for the effluent signal and its location. We hope we have explained the idea better in the revised version.
As for the influence of the temperature and salinity, the use of IOP fractions decreases the necessity of information on water salinity and temperature. We don’t calculate the actual concentrations of the CDOM, detritus and phytoplankton. The T and S enter in the algorithm for calculation of IOPs as averaged values and the formula is not very sensitive to those variables (Section 2.2). Analysing salinity anomalies to observe river plumes or wastewater discharges could be misleading for the Black Sea region, where average salinity is approx. 17psu. Additionally, having events so close to the shoreline, where meteorological conditions such as wind speed and direction, and precipitation intensity, makes direct analysis of T,S anomalies too challenging and unreliable. The other reason not to use T,S information is that we do not have information with such high resolution and the events we want to detect are of very small scale. Our aim was to concentrate on optical Sentinel images.
We have added a paragraph explaining this in the revised version (Section 2.4)
'4. Comparisons with the others’ methods in the literature is necessary and convincing.'
The comment is taken into account. In the Introduction is added a paragraph with more literature references to similar studies. The Section 2.4 is added with a list of found in the literature articles dealing with detection of river plumes and the methods are explained. We also explained the main differences and specifics in our study.
'4. Line 174. “Error! Reference source not found”.
5. Line 195. “Error! Reference source not found”.
6. Line 209. “Error! Reference source not found”.
7. Lines 221,223,226,252…too many “Error! Reference source not found”.'
This was a really unfortunate conversion from the original file. We will ensure that it does not happen again.
Reviewer 3 Report
The paper focuses on an interesting issue. However, there is some improvements needs to be done for publication.
Authors did not provide complete information stressing novelty of their research.
Section “Discussion” need compare obtained results to some other studies in the Black Sea region or elsewhere in the world to support or discuss the author’s opinion.
Lines 102-127: it is more correct to put this information in the “Methods” section
Please, make figures (especially, Figure 4. "Water-leaving...") more evident.
Please, fix links to figures
Author Response
We would like to thank the anonymous reviewer for the dedicated time and the useful comments. Please find our replies to the remarks below.
'The paper focuses on an interesting issue. However, there is some improvements needs to be done for publication.
Authors did not provide complete information stressing novelty of their research.'
The comment is taken into account. The Introduction is revised and a paragraph stating what is the main difference from other studies is added. Mainly, in our study we used the fractions of absorption coefficients due to detritus, CDOM and phytoplankton instead of the concentration. This way we avoid dealing with the different value ranges of the variables at different places. Furthermore, we decrease the uncertainty coming from different bio-optical algorithms to calculate the concentration values and the need to use information on the water physical characteristics (temperature, salinity etc).
'Section “Discussion” need compare obtained results to some other studies in the Black Sea region or elsewhere in the world to support or discuss the author’s opinion.'
The comment is taken into account. The Section 2.4 is added with a list of found in the literature articles dealing with detection of river plumes and the methods are explained. We also explained the main differences and specifics in our study. Then in the Discussions we added the outcomes of other papers and how they correspond to our results.
'Lines 102-127: it is more correct to put this information in the “Methods” section'
The comment is taken into account and the information is moved to Section “Methods”
'Please, make figures (especially, Figure 4. "Water-leaving...") more evident.'
The comment is taken into account: the figures are produced with better quality and increased size of the numbers.
'Please, fix links to figures'
Sorry, we will ensure that this does not happen again after the conversion of the file.
Round 2
Reviewer 2 Report
The author has addressed all my comments.